# Neural Solvers for Fast and Accurate Numerical Optimal Control

**Federico Berto**[*], **Stefano Massaroli**[†], **Michael Poli**[*], **Jinkyoo Park**[*]

## Abstract

Synthesizing optimal controllers for dynamical systems in practice involves solving real–time optimization problems with hard time constraints. These constraints restrict the class of numerical methods that can be applied; indeed, computationally expensive but accurate numerical routines often have to be replaced with fast and inaccurate methods, trading inference time for worse theoretical guarantees on solution accuracy. This paper proposes a novel methodology to accelerate numerical optimization of optimal control policies via *hypersolvers*, hybrids of a base solver and a neural network. In particular, we apply low–order explicit numerical methods for the *ordinary differential equation* (ODE) associated to the numerical optimal control problem, augmented with an additional parametric approximator trained to reduce local truncation errors introduced by the base solver. Given a target system to control, we first pre-train hypersolvers to approximate base solver residuals by sampling plausible control inputs. Then, we use the trained hypersolver to obtain fast and accurate solutions of the target system during optimization of the controller. The performance of our approach is evaluated in direct and model predictive optimal control settings, where we show consistent Pareto improvements in terms of solution accuracy and control performance.

## 1 Introduction

Emerging applications of optimal control to complex, high–dimensional systems require the implementation of computationally expensive numerical methods for *ordinary differential equations* (ODEs) (Wanner & Hairer, 1996; Butcher, 1997). Here, real–time and hardware constraints render certain numerical methods not applicable, forcing the application of computationally cheaper but otherwise less accurate algorithms. While the paradigm of optimal control has successfully been applied in various domains (Vadali et al., 1999; Lewis et al., 2012; Zhang et al., 2016), these computation budget constraints are still a great challenge (Ross & Fahroo, 2006; Baotić et al., 2008). Furthermore, lower–order explicit ODE solvers are known to be brittle in the presence of stiff dynamical systems (Wanner & Hairer, 1996), resulting in numerical issues and unreliable controllers.

This work alleviates these limitations by detailing a systematic procedure for the *offline* optimization and subsequence *online* application of ODE solvers equipped with a neural network approximator of higher–order local truncation residuals. These hybrid solvers, known as *hypersolvers* (Poli et al., 2020a), employ a low–order, computationally cheaper solver while preserving theoretical local and global solution accuracy guarantees of traditional higher order methods. We extend the range of applicability of hypersolvers to controlled dynamical systems, investigating and developing several strategies to enable their application during online solution of the optimal control problem without *finetuning*. Namely, we consider hypersolvers whose neural network residual approximators are *a priori* optimized on an uncontrolled target system and then applied to significantly speed up iterated simulations of the system later required by the control policy optimization algorithm. The proposed

---

[*]KAIST, {`fberto, poli_m, jinkyoo.park`} `@kaist.ac.kr`
[†]The University of Tokyo, `massaroli@robot.t.u-tokyo.ac.jp`

35th Conference on Neural Information Processing Systems (NeurIPS 2021), Sydney, Australia.

two-stage scheme is tested in both direct and model predictive control settings and shows promising Pareto optimal results in terms of solution accuracy and subsequent control performance.

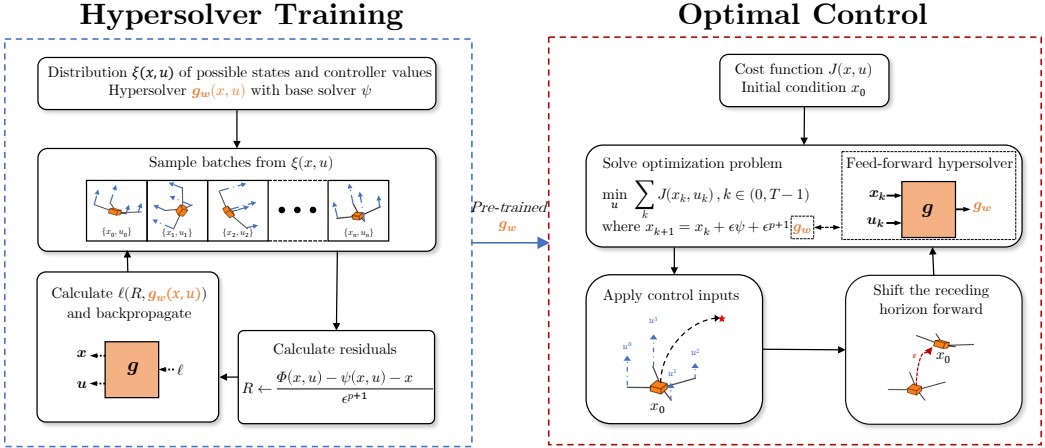

Figure 1: Overview of the proposed method. [Left] The hypersolver is trained to approximate residuals given a distribution of control inputs and states. [Right] The pre–trained hypersolver model is used to accelerate the online learning of a control policy via Model Predictive Control.

## 2 Numerical Optimal Control

Let us consider the following controlled dynamical system described:

$$\dot{x}(t) = f(t, x(t), u_\theta(t))$$
$$x(0) = x_0 \tag{1}$$

with state $x \in \mathcal{X} \subset \mathbb{R}^{n_x}$, input $u_\theta \in \mathcal{U} \subset \mathbb{R}^{n_u}$ defined on a compact time domain $\mathcal{T} := [t_0, T]$ where $\theta$ is a set of free parameters of the controller. Solutions of equation 1 are denoted with $x(t) = \Phi(x(s), s, t)$ for all $s, t \in \mathcal{T}$.

### 2.1 Optimal Control Problem

Given some objective function $J : \mathcal{X} \times \mathcal{U} \to \mathbb{R}; \ x_0, u_\theta \mapsto J(x_0, u_\theta(t))$, and a distribution $\rho_0(x_0)$ of initial conditions with support in $\mathcal{X}$, we consider the following nonlinear program, constrained to the dynamics 1:

$$\min_{u_\theta(t)} \quad \mathbb{E}_{x_0 \sim \rho_0(x_0)} \left[ J(x_0, u_\theta(t)) \right]$$
$$\text{subject to} \quad \dot{x}(t) = f(t, x(t), u_\theta(t))$$
$$x(0) = x_0 \tag{2}$$
$$t \in \mathcal{T}$$

where the controller parameters $\theta$ are optimized. We will omit the use of $\theta$ and write $u(t) = u_\theta(t)$ from now on for simplifying the notation. Since analytic solutions of 2 exist only for limited classes of systems and objectives, numerical solvers are often applied to iteratively find a solution instead: for this reason we refer to problem 2 as *numerical optimal control*.

### 2.2 Solver Residuals for Controlled Dynamical Systems

Given nominal solutions $\Phi$ of equation 1 we can define the *residual* of a numerical ODE solver as the normalized error accumulated in a single step size of the method, i.e.

$$R_k = R(t_k, x(t_k), u(t_k)) = \frac{1}{\epsilon^{p+1}} \left[ \Phi(x(t_k), t_k, t_{k+1}) - x(t_k) - \epsilon \psi_\epsilon(t_k, x(t_k), u(t_k)) \right] \tag{3}$$

where $p$ is the order of the numerical solver corresponding to $\psi_\epsilon$.

# 3 Hypersolvers for Optimal Control

## 3.1 Formulation

Given a $p$–order base solver update map $\psi_\epsilon$, the corresponding *hypersolver* is the discrete iteration

$$x_{k+1} = x_k + \overbrace{\epsilon \psi_\epsilon \left(t_k, x_k, u_k\right)}^{\text{base solver step}} + \epsilon^{p+1} \underbrace{g_\omega \left(t_k, x_k, u_k\right)}_{\text{approximator}} \tag{4}$$

where $g_\omega \left(t_k, x_k, u_k\right)$ is some $o(1)$ parametric function with free parameters $\omega$. The core idea is to select $g_\omega$ as some function with *universal approximation* properties and fit the higher-order terms of the base solver by explicitly minimizing the residuals over a set of state and input samples.

## 3.2 Training

We assume the considered optimal control problems to feature a control input constrained in a set $\mathcal{U}$. Such constraints are often present either due to physical limitations of the actuators or safety restraints of the workspace; while this is not directly enforced nor necessary, the limited dataset of constrained controls ensures that hypersolver pre–training is performed successfully. The analysis considers the case of *time–invariant systems* (A.3) in which dynamics of equation 1 is not directly dependent on time $t$, thus we can rewrite $g_w(t, x(t), u(t)) = g_w(x, u)$. We also check the generalization properties of hypersolvers: details are available in the Appendix B.

The training procedure can be written as the following optimization problem

$$\min_w \quad \mathbb{E}_{(x,u) \sim \xi(x,u)} \| R(\epsilon, x, u) - g_w\left(x, u\right) \|_2 \tag{5}$$

where $\xi(x, u)$ is a distribution with support in $\mathcal{X} \times \mathcal{U}$ of the state and controller spaces. To guarantee sufficient exploration of the state-controller space, we use Monte Carlo sampling (Robert & Casella, 2013) from the given distribution.

In particular, batches of initial conditions $\{x_0^i\}$, $\{u_0^i\}$ are sampled from $\xi$ and the optimization problem of equation 5 is solved via backpropagation for updating the parameters of the hypersolver $g_w$ using a *stochastic gradient descent* (SGD) algorithm, e.g., Adam (Kingma & Ba, 2017) and we repeat the procedure for every training epoch.

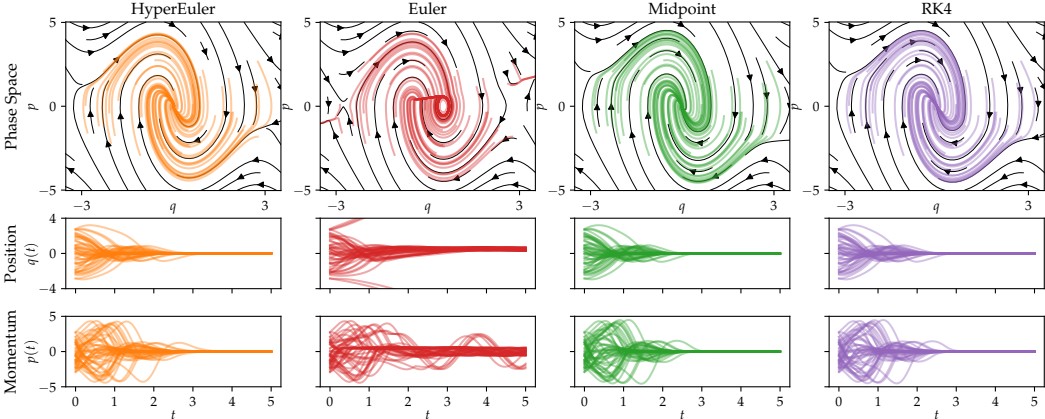

Figure 2: Pendulum trajectories simulated by an accurate adaptive–step solver with controllers optimized via direct optimal control. While the controller optimized with the Euler solver fails to control the system for some trajectories, the one obtained with HyperEuler can improve the performance while introducing a minimal overhead with results comparable to higher-order solvers.

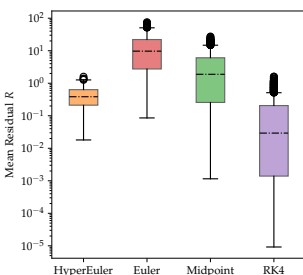 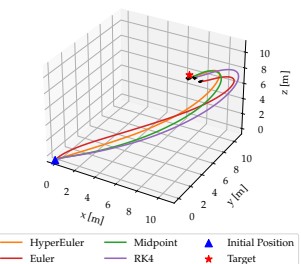 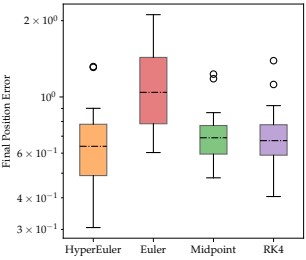

Figure 3: [Left] Local residual distribution for the quadcopter model for step size $\epsilon = 0.02\ s$. [Center] Trajectories of controlled quadcopters with MPC whose receding horizon controller is optimized by solving the ODE with different methods. [Right] Final positions error distribution. The proposed approach with HyperEuler achieves lower average error compared to other baseline solvers, while requiring a low overhead compared to higher–order solvers due to a smaller number of dynamics function evaluations.

### 3.3 Optimal Control Optimization with Hypersolvers

The pre-trained hypersolver model is employed for obtaining solutions to the trajectories of the optimal control problem of equation 2. We then optimize the control policy for minimizing the cost function $J$; we repeat the process iteratively until a solution is found. Figure 1 shows an overview of the proposed two–stage approach consisting in hypersolver pre–training and subsequent optimal control.

## 4 Evaluation

### 4.1 Direct Optimal Control of a Pendulum

We consider the inverted pendulum model with a torsional spring described in equation 9. After pre–training the hypersolver by sampling from the sets of possible states $\mathcal{X}$ and controllers $\mathcal{U}$, we apply direct optimal control to optimize over full unrolled trajectories obtained via different fixed-step solvers. The *hypersolved* version of the Euler scheme, HyperEuler, improves control performance while introducing a minimal overhead as shown in Figure 2. Further details are available in the Appendix C.1.

### 4.2 Model Predictive Control of a Quadcopter

We consider the three-dimensional quadcopter model of equation 10. The controller was optimized online via the Model Predictive Control (MPC) (Garcia et al., 1989) algorithm whose receding horizon was calculated through different standard ODE solvers and our hypersolver approach. Figure 3 shows that HyperEuler reduces mean local residuals even compared to the Midpoint solver, while control performance shows the lowest error on final positions even compared to higher–order solvers. Additional experimental details are available in the Appendix C.2.

## 5 Conclusion

We presented a novel method for obtaining fast and accurate control policies. Hypersolver models were firstly pre–trained on distributions of states and controllers to approximate higher–order residuals of base fixed–step ODE solvers. The obtained models were then employed to improve the accuracy of trajectory solutions over which control policies were optimized. We verified that our method shows consistent improvements in the accuracy of ODE solutions and thus on the quality of control policies optimized through numerical solutions of the system. We envision the proposed approach to benefit the control field and robotics in both simulated and potentially real–world environments by efficiently solving high–dimensional time–continuous problems.

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

# A  Additional Hypersolver Material

## A.1  Bounds on Solution Errors

From the definition of residual in equation 3, we can define the *local truncation error* $e_k := \left\| \epsilon^{p+1} R_k \right\|_2$. Given a $p$-th order explicit solver, we have $e_k = o(\epsilon^{p+1})$ (Butcher, 1997). For hypersolvers, similar error bounds are defined as $e_k \le o(\delta \epsilon^{p+1})$, where $\delta > 0$ depends on the hypersolver training results (Poli et al., 2020a, Theorem 1). This result practically guarantees that if $g_\omega$ is a good approximator for $R$, i.e. $\delta \ll 1$, then the overall local truncation error of the *hypersolved* ODE is significantly reduced.

## A.2  Explicit HyperEuler Formulation

During the experiments, we restricted our analysis to the *hypersolved* version of the Euler scheme, namely `HyperEuler` because of the highest potential speed-up it can achieve: since Euler is a first-order method, it requires the least number of function evaluations (NFE) of the vector field $f$ in equation 1 and yields a second order local truncation error $e_k := \left\| \epsilon^2 R_k \right\|_2$, which is larger than other fixed-step solvers. In particular, the base solver scheme $\psi$ of equation 4 can be written as $\psi_\epsilon (t_k, x_k, u_k) = f (t_k, x_k, u_k)$, which is approximating the next state by adding an evaluation of the vector field multiplied by the step size $\epsilon$. The `HyperEuler` scheme can be explicitly written as

$$\begin{cases} x_{k+1} = x_k + \epsilon f(t_k, x_k, u_k) + \epsilon^2 g_w (t_k, x_k, u_k) \\ x_{k=0} = x_0 \end{cases} \tag{6}$$

while its residual can be written as

$$R\left(x(t_k), u(t_k)\right)) = \Phi(x(t_k), t_k, t_{k+1}) - x(t_k) - \epsilon f(t_k, x_k, u_k) \tag{7}$$

## A.3  Hypersolvers for Time-invariant Systems

A *time-invariant* system with time-invariant controller can be described as following

$$\begin{aligned} \dot{x}(t) &= f(x(t), u(x(t))) \\ x(0) &= x_0 \end{aligned} \tag{8}$$

in which $f$ and $u$ do not explicitly depend on time.

## A.4  Architecture

We designed the hypersolver network $g_w$ as a feed–forward neural network. Table 1 summarizes the parameters used for the considered controlled systems, where activation functions `SoftPlus`: $x \mapsto \log(1 + e^x)$ and `Tanh`: $x \mapsto \frac{e^x - e^{-x}}{e^x + e^{-x}}$.

Table 1: Hyper-parameters for the hypersolver networks in the experiments.

|                        | Inverted Pendulum | Quadcopter |
|------------------------|-------------------|------------|
| Input Layer            | 5                 | 28         |
| Hidden Layer 1         | 32                | 64         |
| Activation Function 1  | Softplus          | Softplus   |
| Hidden Layer 2         | 32                | 64         |
| Activation Function 2  | Tanh              | Softplus   |
| Output Layer           | 2                 | 12         |

We also used the vector field $f$ as an input of the hypersolver, which, however, does not require a further evaluation since it is pre-evaluated at runtime by the base solver $\psi$. We emphasize that the size of the network should depend on the application: a too-large neural network may require more computations than just increasing the numerical solver's complexity: Pareto optimality of hypersolvers also depends on their complexity. By keeping their neural network small enough, we can guarantee that evaluating the hypersolvers is cheaper than resorting to more complex numerical routines.

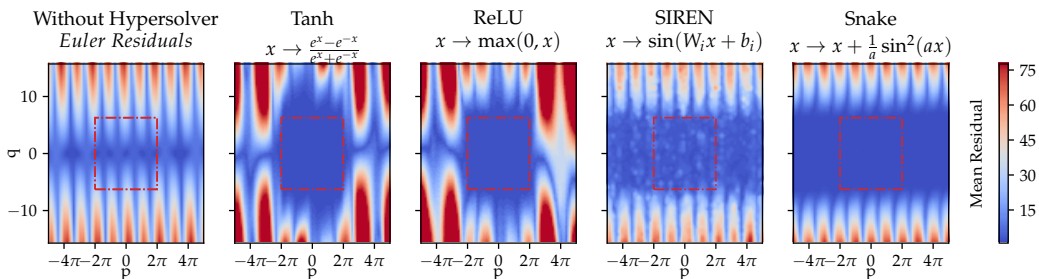

Figure 4: Generalization outside of the training region (red rectangle) of the state space with different hypersolver activation functions. Architectures containing activation functions with periodic components achieve better extrapolation properties compared to the others.

## B  Generalization Study

We assumed the state and controller spaces to be bounded and that training is performed sampling for their known distributions. While this is sufficient for optimal control problems given *a priori* known bounds, we also investigate how the system generalizes to unseen states and control input values.

In particular, we found that activation functions have an impact on the end result of generalization beyond known boundaries. We took into consideration two commonly used activation functions, Tanh : $x \to \frac{e^x - e^{-x}}{e^x + e^{-x}}$ and ReLU : $x \to \max(0, x)$, along with network architectures which employ activation functions containing periodic components: SIREN : $x \to \sin(Wx + b)$ (Sitzmann et al., 2020) and Snake : $x \to x + \frac{1}{a}\sin^2(ax)$ (Ziyin et al., 2020). We trained hypersolver models with the different activation functions for the inverted pendulum of equation 9 with common settings: time step $\epsilon$ was chosen as $0.1\ s$, networks were trained for $10^5$ epochs with the Adam optimizer and learning rate of $10^{-3}$, where sampling was done on from the uniform distribution $\xi(x, u)$ with support in $\mathcal{X} \times \mathcal{U}$ with $\mathcal{X} = [-2\pi, 2\pi] \times [-2\pi, 2\pi]$ and $\mathcal{U} = [-10, 10]$. Figure 4 and Figure 5 show generalization outside the states and controllers distribution respectively. We notice that while Tanh and ReLU perform well on the training set of interest, performance degrades rapidly outside of it. On the other one hand, SIREN and Snake manage to extrapolate the periodicity of the residual distribution even outside of the training region, thus providing further empirical evidence of the universal approximation theorem (Ziyin et al., 2020, Theorem 3).

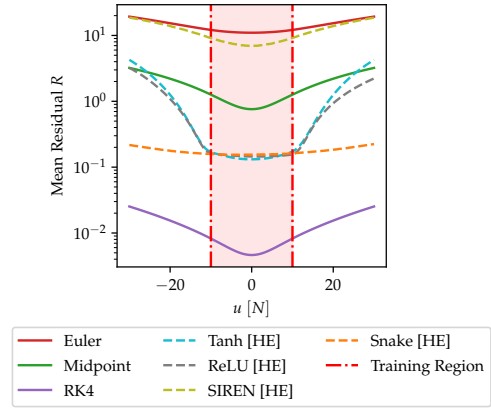

Figure 5: Generalization outside of the training region (red lines) in the controller space with different hypersolver activation functions. The Snake activation function manages to extrapolate best to unseen controllers compared to others.

## C  Experimental Details

### C.1  Inverted Pendulum

**System Dynamics**   We model the inverted pendulum with elastic joint with Hamiltonian dynamics via the following:

$$\begin{bmatrix} \dot{q} \\ \dot{p} \end{bmatrix} = \begin{bmatrix} 0 & 1/m \\ -k & -\beta/m \end{bmatrix} \begin{bmatrix} q \\ p \end{bmatrix} - \begin{bmatrix} 0 \\ mgl\sin q \end{bmatrix} + \begin{bmatrix} 0 \\ 1 \end{bmatrix} u \tag{9}$$

where $m = 1\ [Kg]$, $k = 0.5\ [N/\mathrm{rad}]$, $r = 1\ [m]$, $\beta = 0.01\ [Ns/\mathrm{rad}]$, $g = 9.81\ [m/s^2]$.

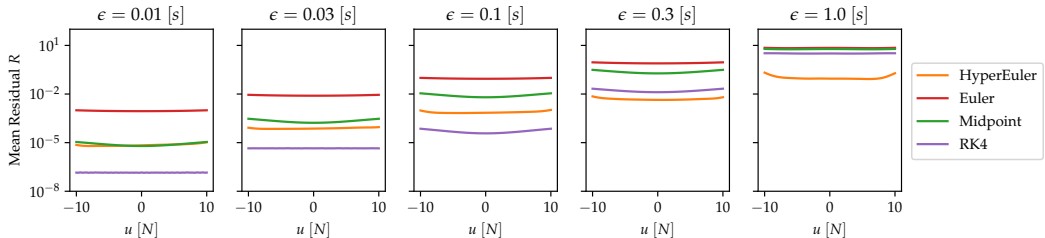

Figure 6: Mean local residuals of the inverted pendulum as a function of control inputs at different step sizes $\epsilon$. HyperEuler improves on the local residuals compared to the baseline Euler and even compared to higher-order ODE solvers at larger step sizes.

**Hypersolver pre–training**   We select $\xi(x,u)$ as a uniform distribution with support in $\mathcal{X} \times \mathcal{U}$ where $\mathcal{X} = [-2\pi, 2\pi] \times [-2\pi, 2\pi]$ and $\mathcal{U} = [-5, 5]$ to guarantee sufficient exploration of the state-controller space. Nominal solutions $\Phi$ are calculated using the `Tsitouras 5` (Tsitouras, 2011) adaptive–step solver with absolute and relative tolerances set to $10^{-5}$. We train the hypersolver on local residuals using the `Adam` optimizer with learning rate of $3 \times 10^{-4}$ for $3 \times 10^5$ epochs. Figure 6 shows mean local residuals with hypersolver models trained on different step sizes.

**Direct optimal control**   The goal is to stabilize the inverted pendulum in the vertical position $x^\star = (0,0)$. We choose $t \in [0,3]$ and a step size $\epsilon = 0.2\ s$ for the experiment. The control input is assumed continuously time–varying. The neural controller is optimized via SGD with `Adam` with learning rate of $3 \times 10^{-3}$ for 1000 epochs. Trajectories obtained with the controller optimized with HyperEuler reach final positions $q = (1.6 \pm 17.6) \times 10^{-2}$ while Midpoint and RK4 ones $q = (-0.6 \pm 12.7) \times 10^{-2}$ and $q = (1.1 \pm 12.8) \times 10^{-2}$ respectively. On the other hand, the controller optimized with the Euler solver fails to control some trajectories obtaining a final $q = (6.6 \pm 19.4) \times 10^{-1}$. HyperEuler considerably improved on the Euler baseline while requiring only 1.2% more *Floating Point Operations* (FLOPs) and 49.5% less compared to Midpoint.

## C.2   Quadcopter

**System Dynamics**   The quadcopter model is a suitably modified version of the explicit dynamics update in (Panerati et al., 2021) for batched training in PyTorch. The dynamic model is described by the following accelerations update:

$$
\begin{aligned}
\ddot{\mathbf{x}} &= \left( \mathbf{R} \cdot [0, 0, k_F \textstyle\sum_{i=0}^{3} \omega_i^2] - [0, 0, mg] \right) m^{-1} \\
\ddot{\boldsymbol{\psi}} &= \mathbf{J}^{-1} \left( \tau(l, k_F, k_T, [\omega_0^2, \omega_1^2, \omega_2^2, \omega_3^2]) - \dot{\boldsymbol{\psi}} \times \left( \mathbf{J}\dot{\boldsymbol{\psi}} \right) \right)
\end{aligned}
\tag{10}
$$

where $\mathbf{x} = [x, y, z]$ corresponds to the drone positions and $\boldsymbol{\psi} = [\phi, \theta, \psi]$ to its angular positions; $\boldsymbol{R}$ and $\boldsymbol{J}$ are its rotation and inertial matrices respectively, $\tau(\cdot)$ is a function calculating the torques induced by the motor speeds $\omega_i$, while arm length $l$, mass $m$, gravity acceleration constant $g$ along with $k_F$ and $k_T$ are scalar variables describing the quadcopter's physical properties.

**Hypersolver pre–training**   We select $\xi(x,u)$ as a uniform distribution with support in $\mathcal{X} \times \mathcal{U}$ where $\mathcal{X}$ is chosen as a distribution of possible visited states and each of the four motors $i \in [0,3]$ has control inputs $u^i \in [0, 2.17] \times 10^5$ `rpm`. Nominal solutions $\Phi$ are calculated using the `Dormand-Prince 5` (Dormand & Prince, 1980) adaptive–step solver with relative and absolute tolerances set to $10^{-7}$ and $10^{-9}$ respectively. We train HyperEuler on local residuals using the `Adam` optimizer with learning rate of $10^{-3}$ for $10^5$ epochs.

**Model predictive control**   The control goal is to reach a final positions $(x, y, z)^\star = (8, 8, 8)\ m$. We choose $t \in [0,3]$ and a step size $\epsilon = 0.02\ s$ for the experiment. The control input is assumed piece–wise constant during MPC sampling times. The receding horizon is chosen as $0.5\ s$. The neural controller is optimized via SGD with `Adam` with learning rate of $10^{-2}$ for 20 iterations at

each sampling time. 30 experiments are conducted starting at random initial conditions which are kept common for the different ODE solvers. HyperEuler requires a single function evaluation per step as for the Euler solver compared to two function evaluations per step for Midpoint and four for RK4. Controlled trajectories optimized with Euler, Midpoint and RK4 collect an error on final positions of $(1.09 \pm 0.37)$ $m$, $(0.71 \pm 0.17)$ $m$, $(0.70 \pm 0.19)$ $m$ respectively while HyperEuler achieves the lowest terminal error value of $(0.66 \pm 0.24)$ $m$.

### C.3 Hardware and Software

Experiments were carried out on a machine equipped with an AMD RYZEN THREADRIP-PER 3960X CPU and two NVIDIA RTX 3090 graphic cards. Software–wise, we used `PyTorch` (Paszke et al., 2019) for deep learning and the `torchdyn` (Poli et al., 2020b) and `torchdiffeq` (Chen et al., 2019) libraries for ODE solvers.

## D Broader Impact

We envision our hypersolver approach to optimal control to be applicable to real–world online control problems. Let us present some considerations that should be taken into account in these regards. Firstly, implementation–wise, *ad–hoc* hardware choices may need to be done for fast inference of the approximator depending on its nature, e.g., neural networks (Sze et al., 2017); additionally, depending on the computational requirements of the control algorithms (Torrente et al., 2021). Secondly, unlike in simulated environments, imperfections in the modeling itself or disturbances can lead to sub–optimal controllers by incorrectly solving the control problem's associated ODEs. However, correction terms can be learned and combined with the dynamics model to improve the control policy (Carron et al., 2019; Hewing et al., 2020). Similarly, hypersolver models trained on residuals derived from nominal data–driven trajectories may be able to correct modeling errors and disturbances to a certain extent. Nonetheless, we believe future works on hypersolvers for optimal control represent an exciting avenue for obtaining fast and accurate optimal control policies not only in simulated, but also in real–world and production environments.

