# OpenReview forum: "Neural Solvers for Fast and Accurate Numerical Optimal Control"
_NeurIPS.cc/2021/Workshop/DLDE — DLDE Workshop -- NeurIPS 2021 Spotlight_

### Official Review · Reviewer_VuZ5 · 2021-09-29
**Review for Neural Solvers for Fast and Accurate Numerical Optimal Control**

**Confidence:** 3

**Review:**

# Summary

This paper proposes a new method to accelerate the optimisation of control policies for ODE systems. Specifically, they train a neural network to correct for residual errors in the numerical ODE solver used during the optimisation of control policies. The network is trained offline, using a training dataset consisting of many different control and state data points, and then applied within each step of the ODE solver during optimisation of the control policy to correct for local errors in the ODE solver state updates. Their main claim is that this improves the accuracy of the ODE solver and hence final control solution, whilst only adding a minimal computational overhead.

# Main review

## Originality

The paper builds on Poli et al 2020, who define the method used in this paper for using a neural network to correct for residuals in the ODE solver. Therefore, the main novelty appears to be the application and investigation of this method for optimising control policies for the ODE system.

## Quality

The mathematical description of the method appears sound. However, there are a few methodological details I do not feel are adequately explained in the paper, which I have detailed below.
1)	How is the “nominal solution” or exact solution determined in order to train the network (i.e. evaluate the residual (equation 3))? This is not mentioned for either of the case studies. Do these case studies have analytical solutions which are used?
2)	What are the actual objective functions used for the control problems (equation 2) for both of the case studies, these are not mentioned? Furthermore, how is this objective function minimised, is it e.g. via gradient descent? In which case, are the gradients of the neural network required in any way for this minimisation?
3)	The second case study considers a dynamical model which includes second order derivatives. It is not fully clear to me how the same Euler ODE solvers + hypersolver method described in the paper for a first order ODE system (equation 2) are transferred to this system?

## Limitations

I am wondering how the step size in the ODE solver affects the results – is the hypersolver more/less useful for larger step sizes? Furthermore, it appears accuracy metrics are only reported using one control target state in each case studied. Averaged metrics over different target states / control tasks would be more robust.

The authors include a generalisation study which is appreciated for understanding the limits of the method.

## Clarity

As above, I think some methodological details could be explained better. Some of the figure numbers in the main text do not appear to match correctly with the figures. Otherwise the paper is well structured.

## Significance

This approach does appear to improve on using standard numerical solvers when solving control problems, but it is not clear to me from the text whether this method has the potential to achieve the high-level goals stated in the introduction, i.e. accurate, real-time control for real-world problems, some more discussion on this would be useful.


**Score:**

3: Good paper

---

### Official Review · Reviewer_bxG1 · 2021-09-30
**Review for paper number 32 for DLDE Workshop -- NeurIPS 2021**

**Confidence:** 2

**Review:**

This paper proposes a novel methodology to accelerate numerical optimization of optimal control policies via hypersolvers, hybrids of a base solver and a neural network.

Suitable for this workshop. However, it is not clear whether the method works for real-world problems or production environment.

**Score:**

3: Good paper

---

### Official Review · Reviewer_zAqg · 2021-10-11
**Review for**

**Confidence:** 3

**Review:**

Summary

The paper aims to show how hypersolvers can be used for controlled dynamical systems.
The paper is within the scope of the workshop but it would benefit from revisions (some typos in Fig and text).

Comments

The authors explain the approach in some detail, but the explanation of the methods and the results could be clearer (e.g. lines 67-69, Fig 1, there is no accompanying text for Fig 2, and 3.1), How does Fig 3.1 relate to Fig 3.2?
The paper would also benefit from some connection between how the experimental results support the original aim that this approach can be successfully used online.


**Score:**

3: Good paper

---

### Decision · Program_Chairs · 2021-10-15

**Decision:**

Accept (Spotlight)

**Comment:**

Reviewers were very consistent and recommended acceptance. Authors may wish to address the reviewers’ concerns about clarity. Additionally, all reviewers were curious to hear more about how this work might be extended to practical/real-world applications. Hypersolvers are a major application of DLDE, and this will make a good spotlight for the workshop.